# Maintenance Treatment for Metastatic Pancreatic Cancer: Balancing Therapeutic Intensity with Tolerable Toxicity

**DOI:** 10.3390/cancers15143657

**Published:** 2023-07-18

**Authors:** Evan J. Walker, Andrew H. Ko

**Affiliations:** 1Division of Hematology and Oncology, Department of Medicine, University of California, San Francisco, CA 94158, USA; 2Helen Diller Comprehensive Cancer Center, University of California, San Francisco, CA 94158, USA

**Keywords:** pancreatic cancer, maintenance, chemotherapy, immunotherapy, PARP inhibitor, quality of life

## Abstract

**Simple Summary:**

Metastatic pancreatic cancer is conventionally treated with chemotherapy until the development of either progressive disease or intolerable cumulative toxicity. Maintenance treatment refers to the concept of de-escalating therapy, either by paring back on the components of multi-agent chemotherapy or by the introduction of novel agents, in patients who have achieved a favorable response (stable disease or better) following a period of first-line chemotherapy. The goal of maintenance treatment is to maintain disease control and maximize progression-free intervals while preserving or improving patients’ quality of life and minimizing cumulative toxicity. Here, we summarize the existing data to guide the use of maintenance treatments and discuss future directions in the field.

**Abstract:**

Metastatic pancreatic ductal adenocarcinoma is typically treated with multi-agent chemotherapy until disease progression or intolerable cumulative toxicity. For patients whose disease shows ongoing control or response beyond a certain timeframe (≥3–4 months), options include pausing chemotherapy with close monitoring or de-escalating to maintenance therapy with the goal of prolonging progression-free and overall survival while preserving quality of life. There is currently no universally accepted standard of care and a relative dearth of randomized clinical trials in the maintenance setting. Conceptually, such therapy can entail continuing the least toxic components of a first-line regimen and/or introducing novel agent(s) such as the poly(ADP-ribose) polymerase inhibitor olaparib, which is presently the only approved drug for maintenance treatment and is limited to a genetically defined subset of patients. In addition to identifying new therapeutic candidates and combinations in the maintenance setting, including targeted agents and immunotherapies, future research should focus on better understanding this unique biologic niche and how treatment in the maintenance setting may be distinct from resistant/refractory disease; identifying molecular predictors for more effective pairing of specific treatments with patients most likely to benefit; and establishing patient-reported outcomes in clinical trials to ensure accurate capture of quality of life metrics.

## 1. Introduction

Pancreatic ductal adenocarcinoma (PDAC) will soon become the second-leading cause of cancer-related mortality in the US [1,2]. Incidence and death rates are rising, with over 62,000 new diagnoses and roughly 50,000 deaths annually [3]. Up to 85% of PDAC cases are diagnosed at an advanced, incurable stage, and the five-year survival rate for metastatic disease is only 3% [3]. 

Cytotoxic chemotherapy is the mainstay of treatment for advanced PDAC. For patients who can tolerate aggressive therapy, phase III data support the use of combination regimens such as 5-fluorouracil (5-FU), irinotecan, and oxaliplatin (FOLFIRINOX) [4] or gemcitabine and nano albumin-bound (nab)-paclitaxel [5] as optimal first-line treatments. First-line chemotherapy is generally continued until disease progression, with patients often reaching a plateau in response after 4–6 months. However, these combination regimens are associated with cumulative toxicity, necessitating frequent dose reductions and/or discontinuation of some or all chemotherapeutic agents prior to tumor progression. For example, in the PRODIGE 4/ACCORD 11 trial establishing FOLFIRINOX as a standard of care, patients received a median of 10 cycles (5 months), and in practice, this regimen is conventionally limited to 12 cycles (6 months) before de-escalating therapy or moving to a treatment hiatus to alleviate toxicity.

For patients with responsive or stable disease after 4–6 months of first-line therapy, therapeutic options exist on a spectrum of aggressiveness between indefinite continuation of first-line therapy and complete cessation of treatment (a chemotherapy “holiday”), and there is a tradeoff between maintaining optimal disease control and the ongoing and cumulative toxicity associated with treatment. Maintenance therapy, either with the continuation of components of first-line regimens (often in some attenuated fashion) or the introduction of novel agent(s), attempts to balance these goals by preventing or delaying disease progression, forestalling the need for second-line chemotherapy, and minimizing treatment-related toxicity to preserve or enhance quality of life. Maintenance therapy can be effective if toxicity is minimized without compromising overall survival (OS) (See Figure 1A,B); the ideal maintenance therapy achieves concurrent reduction of toxicity with actual prolongation of survival (Figure 1C).

The concept of maintenance therapy is well described in the treatment of other solid tumors. For example, in ovarian cancer, the combination of the anti-vascular endothelial growth factor (VEGF) antibody bevacizumab and the poly(ADP-ribose) polymerase (PARP) inhibitor olaparib [6], as well as the PARP inhibitor niraparib [7], prolongs progression-free survival (PFS) after initial treatment with platinum-based chemotherapy. In lung cancer, multi-agent first-line chemotherapy regimens are often simplified to constituent monotherapies such as docetaxel [8], gemcitabine [9], or pemetrexed [10]. In colon cancer, first-line fluoropyrimidine, oxaliplatin, and bevacizumab are often followed by fluoropyrimidine/bevacizumab maintenance [11,12]. However, there is limited prospective data to inform maintenance treatment decisions for metastatic PDAC. Herein, we review the available data, ongoing studies, and best practices for future advancements in this field. Of note, we are defining maintenance therapy here as additional *systemic* treatment that is administered to patients who have achieved disease control or response following induction chemotherapy. We recognize there is also considerable interest in locoregional approaches, such as stereotactic body radiation (SBRT), to treat patients with oligometastatic PDAC who have shown a favorable response to initial systemic treatment, and while this has created compelling opportunities for novel sequencing of different treatment modalities and represents the subject of ongoing research [13,14], it is beyond the scope of this review. 

## 2. Continuation Maintenance Therapy

The most straightforward maintenance treatment strategy involves the simplification of first-line multi-agent chemotherapeutic regimens. For example, the triplet FOLFIRINOX can be reduced to the doublet of 5-FU/oxaliplatin (FOLFOX), the doublet of 5-FU/irinotecan (FOLFIRI), or fluoropyrimidine monotherapy, eliminating the chemotherapeutic(s) generating the most cumulative toxicity. This strategy is intuitive: patients can make informed decisions based on their direct experience receiving the agents in question, and because maintenance therapy is a subset of the original first-line regimen, therapeutic resistance is minimized and additional treatment options are reserved for later lines. Dose attenuation and/or spacing out treatment intervals can also be incorporated into such a maintenance strategy.

Fluoropyrimidine monotherapy is a minimally toxic option for continuation maintenance after fluoropyrimidine-based first-line combination chemotherapy. This strategy is best supported by the randomized phase II PANOPTIMOX-PRODIGE 35 trial, in which 276 patients were randomized to one of three treatment strategies: FOLFIRINOX for 6 months (Arm A), FOLFIRINOX for 4 months followed by 5-FU maintenance (Arm B), or a sequential strategy alternating gemcitabine and FOLFIRI (Arm C) [15]. This study was conducted to select the best therapeutic option for a subsequent phase III study, and the design precluded a direct statistical comparison of the arms. However, Arms A and B generated numerically similar median PFS (6.3 vs. 5.7 months) and median OS (10.1 vs. 11.2 months). Maintenance therapy in Arm B lasted for a mean of 5.1 months and was followed by the reintroduction of FOLFIRINOX in 30/52 (58%) patients. Counterintuitively, patients in Arm B had increased rates of severe neuropathy (Arm A: 10.2%, Arm B: 19.8%), a finding attributed to a higher median dose intensity of total oxaliplatin exposure in Arm B. Importantly, and despite this discrepancy in neuropathy, the time to deterioration of quality of life was notably longer in the maintenance arm (Arm A: 7.2 months, Arm B: 11.4 months). This study supports the concept that PDAC maintenance therapy can improve quality of life without compromising survival, although confirmatory phase III data are needed. 

Fluoropyrimidine monotherapy as maintenance treatment was also investigated in a single-arm phase II study conducted in China, where the oral fluoropyrimidine S-1 is commonly used [16]. In this prospective study, 32 patients with advanced PDAC were treated with nab-paclitaxel and S-1 for six 3-week cycles, followed by S-1 monotherapy. While the primary objective was to evaluate the combination regimen as first-line therapy for advanced PDAC, results in the subset of patients receiving S-1 maintenance (n = 13) are also instructive, with the median time of treatment on S-1 maintenance of 3 months (range: 1–6 months). Notably, the median OS in the entire study cohort was an impressive 13.6 months, indicating that therapeutic efficacy can be achieved with a relatively modest duration (18 weeks) of intensive induction chemotherapy. 

Several retrospective studies also support fluoropyrimidine maintenance therapy for PDAC. In France, Renure and colleagues reported a cohort of 30 patients with metastatic disease who received capecitabine after 4–8 cycles of FOLFIRINOX, for whom the median duration between initiation of capecitabine and disease progression was 5 months [17]. The median OS was 17 months, a remarkable outcome for metastatic disease, although it should be recognized that this was a selected group with FOLFIRINOX-responsive disease, portending a favorable prognosis. A German group subsequently described their retrospective experience with infusional 5-FU maintenance monotherapy among 13 patients with unresectable PDAC, reporting a median time on maintenance treatment of 14.1 weeks and a median OS of 18.3 months [18]. Taken together, these data suggest capecitabine and infusional 5-FU are likely equivalent as maintenance fluoropyrimidine monotherapy.

It is uncertain whether continuing with a doublet regimen after disease control with FOLFIRINOX offers any therapeutic advantage over maintenance fluoropyrimidine alone. Chevalier et al. reported a multicenter French cohort of 147 patients for whom FOLFIRINOX was de-escalated after at least four cycles to either FOLFIRI (45%), 5-FU monotherapy (35%), FOLFOX (17%), or experimental agents (3%) [19]. Interestingly, maintenance FOLFIRI and 5-FU monotherapy generated similar results (median PFS 9.0 vs. 10.1 months, median OS 18.7 vs. 16.6 months) despite the FOLFIRI group having a better initial performance status. Both groups statistically significantly outperformed FOLFOX (median PFS 6.7 months and median OS 11.8 months). While survival outcomes between FOLFIRI and 5-FU were similar, the FOLFIRI group experienced greater toxicity: 41% of patients developed Grade 3–4 toxicity versus 22% of patients treated with 5-FU alone (*p* = 0.03). This observation, while not definitive, suggests that de-escalation of FOLFIRINOX to FOLFIRI may worsen quality of life with no incremental benefit over 5-FU monotherapy. Conversely, a small German series of 22 patients receiving maintenance FOLFIRI after induction FOLFIRINOX reported Grade 3–4 toxicity in only 18% of patients while generating a promising median PFS of 11 months [20]. The discrepancies in these retrospective reports are likely the result of small numbers and selection bias, as maintenance therapy was chosen at the discretion of treating physicians. At this time, the optimal post-FOLFIRINOX maintenance regimen remains undetermined; patient-specific characteristics and personal preferences alter the risk-benefit balance, and the optimal choice may differ between patients.

For patients receiving first-line treatment with the combination of gemcitabine and nab-paclitaxel, the most commonly employed approach for maintenance therapy is to continue with gemcitabine as monotherapy, as the efficacy of this agent by itself has already been established for the treatment of PDAC [21]. Relias et al. first reported a retrospective analysis of patients treated with a “stop-and-go” strategy of gemcitabine and nab-paclitaxel: for patients who developed Grade 3 neuropathy, nab-paclitaxel was held and then restarted at the time of progression. Among the seven patients who received maintenance gemcitabine monotherapy, the duration of maintenance therapy was 2.8 months, and six patients were able to tolerate the reintroduction of nab-paclitaxel. This strategy was then evaluated prospectively in older (>70 years old) patients with advanced PDAC: patients received induction gemcitabine and nab-paclitaxel for three cycles, followed by pre-planned maintenance gemcitabine if their disease remained controlled after induction [22]. Of 36 total patients, 31 received maintenance gemcitabine for a median of three cycles, resulting in a six-month disease control rate (DCR) of 61%, a median PFS of 6.4 months, and a median OS of 13.4 months. Importantly, this strategy avoided Grade 3 neuropathy in all patients. These encouraging results indicate that, among older patients, treatment efficacy can potentially be preserved and excess toxicity avoided with a planned early switch from gemcitabine and nab-paclitaxel first-line therapy to gemcitabine maintenance. These observational data warrant further evaluation in prospective randomized trials.

While continuation maintenance therapy usually refers to the retention of constituent drugs from a multi-agent induction regimen, an alternative approach involves the introduction of an entirely new cytotoxic regimen to forestall therapeutic resistance and generate a more durable tumor response. In the phase II SEQUENCE trial, presented by Carrato et al. at the 2022 American Society of Clinical Oncology (ASCO) annual meeting, patients with untreated metastatic PDAC received either gemcitabine and nab-paclitaxel or an alternating regimen of cycles of gemcitabine and nab-paclitaxel interspersed with cycles of FOLFOX [23]. The alternating strategy showed an impressive median OS of 13.2 months with statistically significant improvement over the control arm, with a hazard ratio (HR) of 0.68 (95% CI: 0.48–0.95; *p* = 0.023). The contextualization of these results is unclear, particularly when compared with induction of FOLFIRINOX, as is the prediction of resistance to post-progression treatments. Additional prospective study of this strategy is certainly warranted. 

## 3. Novel Agents as Maintenance Therapy

The maintenance setting is a unique niche in cancer treatment after a patient’s disease has been controlled and, ideally, optimally cytoreduced by induction therapy. This allows for the introduction of novel, minimally toxic agents, which alone may not generate a response in untreated PDAC but aim to maintain disease stability once it has been established. In the past decade, several biologic and targeted agents have been tested in this setting, often in genetically defined subsets of PDAC.

The POLO trial is the only completed phase III trial investigating maintenance therapy for patients with metastatic PDAC. In this global randomized study, 154 patients with a confirmed germline mutation in the homologous recombination repair (HRR) genes *BRCA1* or *BRCA2* who had achieved an objective response or stable disease following ≥16 weeks of platinum-based chemotherapy were randomized to receive maintenance treatment with either the PARP inhibitor olaparib or placebo [24]. Mechanistically, tumors deficient in DNA HRR are uniquely sensitive to PARP inhibitors due to synthetic lethality: synergism between the endogenous and pharmacologically-induced blockade of DNA damage repair leads to cell death [25]. This deficiency in DNA damage repair also portends susceptibility to platinum chemotherapeutics [26,27,28], providing a biologic selection mechanism (response/stability with first-line platinum-based chemotherapy) for the use of PARP inhibitors in this maintenance setting. 

Among this selected patient population in the POLO trial, olaparib increased PFS from 3.8 to 7.4 months, with a HR for disease progression or death of 0.53 (95% CI: 0.35–0.82; *p* = 0.004). While this trial met its primary endpoint of PFS improvement, final data analysis revealed no difference in OS (median OS 19.0 months vs. 19.2 months for olaparib vs. placebo, respectively; HR 0.83, *p* = 0.35), although a higher proportion of patients receiving olaparib were alive at 3 years compared with placebo (33.9 vs. 17.8%) [29]. Of note, 27% of patients in the placebo arm subsequently received a PARP inhibitor after disease progression, perhaps explaining in part this lack of an OS benefit. Other secondary outcomes of potential clinical relevance, including time from randomization to second disease progression or death (PFS2), as well as time to initiation of first (TFST) or second (TSST) subsequent therapies following treatment discontinuation or death, favored the olaparib arm. Moreover, although grade ≥ 3 adverse events were not surprisingly observed at a greater frequency in olaparib-treated patients (40% vs. 23%), this did not translate into any difference in health-related quality of life between the two groups [30]. Data on the cost-effectiveness of an olaparib maintenance strategy are mixed, with one analysis reporting it to be cost-effective in the US and China [31] and another in China suggesting it not to be worthwhile [32]. Regardless, in 2019, the US Food and Drug Administration approved olaparib in this setting [33], and it is now recommended as a preferred maintenance regimen for patients with germline *BRCA1/2* mutations in the National Comprehensive Cancer Network guidelines [34]. Importantly, since the POLO trial only compared PARP inhibition to placebo, it remains unknown whether there is an incremental benefit of PARP inhibition compared with other strategies, such as continuation maintenance therapy (e.g., capecitabine).

While the POLO trial was a positive study by virtue of meeting its primary PFS endpoint, the results are only generalizable to the <5% of patients harboring germline *BRCA1/2* mutations [35,36]. Subsequent research has attempted to better define and possibly broaden the population that might benefit from DNA damage response inhibition in this context. Mutations in other genes besides *BRCA1/2* that are also key components of HRR machinery, for example, may also confer susceptibility to PARP inhibitors. A single-arm phase II study by Reiss and colleagues evaluated the PARP inhibitor rucaparib as maintenance therapy for patients with locally advanced or metastatic PDAC [37], using a similar definition for platinum sensitivity as the POLO trial (i.e., stable or responsive disease following ≥16 weeks of platinum-based induction chemotherapy), but broadening the eligibility criteria to include patients with either germline or somatic pathogenic variants in *BRCA1, BRCA2,* or *PALB2*. Results from this 46-patient study were quite impressive, with an observed median PFS and OS of 13.1 months and 23.5 months, respectively. Of 36 patients with measurable disease, 42% experienced a disease response. Importantly, these data built upon the findings of POLO, with responses noted in three patients with germline *PALB2* mutations and one patient with a somatic *BRCA2* mutation. Preclinical data using an *ATM*-deficient PDAC mouse model [38] suggest a maintenance strategy targeting DNA damage repair may be applicable and potentially effective in this genetic subset as well. Taken together, these findings support expanding the role of PARP inhibitors beyond the group of patients with germline *BRCA1/2* mutations to cancers that harbor other DNA repair deficiencies, both germline and somatic. However, additional studies are needed to define with greater precision the subset of patients who will benefit from PARP inhibitors. Recent advances in parallel germline/somatic genomic classification techniques, using whole genome sequencing to define HRR-deficient PDAC instead of grouping tumors based on individual mutations, may provide the key to accurately predicting PARP inhibitor sensitivity [39].

Combining PARP inhibitors with potentially synergistic novel agents represents another compelling strategy to expand the group of patients who might benefit from DNA damage response inhibition. In pre-clinical models, PARP inhibition upregulates the expression of programmed death ligand 1 (PD-L1) on the surface of cancer cells and synergistically enhances the activity of immune checkpoint inhibitors targeting PD-(L)1 [40,41], providing a biologic rationale for PARP/PD-(L)1 dual inhibition. Similarly, in mouse models of BRCA1-deficient ovarian cancer, PARP inhibition and immune checkpoint inhibition targeting cytotoxic T-lymphocyte-associated protein 4 (CTLA-4) increased T-cell tumor infiltration and interferon-gamma production, improving survival [42,43]. Reiss and colleagues recently conducted a phase Ib/II trial to evaluate this strategy in PDAC [44]. In this randomized study, non-biomarker-selected patients with advanced PDAC who had stable/responding disease after 4 months of platinum-based chemotherapy received maintenance therapy with the PARP inhibitor niraparib in combination with one of two immune checkpoint inhibitors: either the anti-programmed cell death protein-1 (PD-1) antibody nivolumab (n = 44) or the anti-CTLA-4 antibody ipilimumab (n = 40). The primary endpoint was PFS at 6 months. While patients on the niraparib/nivolumab arm had only a modest median PFS of 1.9 months and a 6-month PFS rate of 20.6%, the niraparib/ipilimumab arm met the primary endpoint: median PFS was 8.1 months and PFS at 6 months was 59.6% (95% CI: 44.3–74.9%, *p* = 0.045), exceeding the predefined 6-month PFS threshold of 44%. The median OS, meanwhile, was an impressive 17.3 months in this cohort. Notably, in the niraparib/ipilimumab arm, 33 (82.5%) patients were germline *BRCA1/2* and *PALB2* wild-type, and 30 (75%) patients had no germline DNA HRR variant detected of any kind. Among these 30 patients without HRR variants, the median PFS was still 7.6 months, and the median OS was 15 months. This study provides the most promising evidence to date for a non-cytotoxic maintenance therapy that can benefit patients selected not based on genomic characteristics but solely based on the platinum-responsive nature of their tumors. It also provides exciting evidence for the use of immunotherapy for PDAC, a tumor historically resistant to this treatment. 

While PARP inhibitors are the most promising and well-studied therapeutic agents to date in this maintenance setting, other non-cytotoxic agents have also been explored in this clinical context in modest-sized trials. Reni and colleagues reported a randomized phase II trial of 56 patients comparing the multitargeted tyrosine kinase inhibitor sunitinib with observation after 6 months of induction chemotherapy [45], with a primary outcome measure of PFS at 6 months. The primary endpoint was achieved, with 6-month PFS rates of 22.2 vs. 3.6% (*p* < 0.01) in favor of sunitinib-treated patients. Additionally, 2-year OS was also improved in the sunitinib arm (22.9% vs. 7.1%), although median OS did not significantly differ (10.6 vs. 9.2 months). Main grade 3–4 adverse events associated with sunitinib included cytopenias, diarrhea, and hand-foot syndrome. To date, no confirmatory phase III trial has been conducted, and sunitinib cannot be recommended as PDAC maintenance therapy. More recently, Bever and colleagues reported results from a small phase Ib study of the biguanide antidiabetic drug metformin with or without the mammalian target of rapamycin (mTOR) inhibitor rapamycin as maintenance treatment after ≥6 months of chemotherapy [46]. Among 22 patients (11 receiving metformin and 11 receiving the combination), the median PFS was 3.5 months and the median OS was 13.2 months, with 37% of patients remaining alive at 2 years. The addition of rapamycin generated no significant differences in outcomes but increased the rate of treatment-related adverse events from 0% to 27%. While these represent encouraging early findings, much further prospective data are needed before considering the use of metformin in the maintenance setting. Finally, erlotinib, a small-molecule inhibitor of the epidermal growth factor receptor (EGFR), was tested as maintenance therapy for locally advanced PDAC in the LAP07 trial [47]. In this complex study, patients were first randomized to an induction regimen of gemcitabine with or without erlotinib, and then those with controlled disease after 4 months of treatment were again randomized to consolidative chemoradiation or 2 additional months of systemic therapy. Patients in the initial gemcitabine/erlotinib arm without progressive disease subsequently received maintenance erlotinib. While the primary objective of the study was to evaluate the effect of consolidative chemoradiation after 4 months of induction chemotherapy, the gemcitabine arm was also compared with the gemcitabine/erlotinib arm, providing an indirect evaluation of the efficacy of maintenance EGFR inhibition. Unfortunately, the combination/maintenance arm trended towards worse overall survival, with a median OS of 11.9 months vs. 13.6 months (HR 1.19, 95% CI: 0.97–1.45, *p* = 0.09). Based on these and other data, EGFR inhibition is not routinely used in PDAC treatment. 

Vaccine-based immune modulation has also been explored as a therapeutic avenue for PDAC maintenance. In a phase II study, Wu and colleagues randomized patients with responding or stable metastatic PDAC after 8–12 cycles of FOLFIRINOX to either continuation of chemotherapy or an allogeneic granulocyte-macrophage colony-stimulating factor-transfected pancreatic tumor vaccine given with ipilimumab [48]. While correlative studies showed the vaccine modulated T-cell differentiation and increased tumoral M1 macrophages, the vaccine arm demonstrated worse survival (median OS 14.7 versus 9.4 months), and the study was stopped for futility at interim analysis. At the 2022 ASCO annual meeting, Hilmi and colleagues presented interim results from the randomized phase II TEDOPAM GERCOR D17-01 PRODIGE 63 study testing maintenance therapy with the novel cancer vaccine OSE2101 [49]. This vaccine, which is restricted to patients with a Human Leukocyte Antigen (HLA)-A2 genotype, targets five tumor-associated antigens (CEA, HER2, MAGE2, MAGE3, and TP53). The study was originally designed to include three arms in which patients with advanced PDAC who had achieved an objective response or disease stabilization following 8 weeks of FOLFIRINOX were randomized to receive continuation maintenance therapy with FOLFIRI, OSE2101 alone, or OSE2101 in combination with nivolumab. However, tumor flares in the third arm prompted the redesign of the trial to include a comparison between FOLFIRI and FOLFIRI/OSE2101. Thus far, early data have been presented from the FOLFIRI (n = 9) and the original OSE2101 monotherapy (n = 10) arms, showing median OS of 11.5 and 9.6 months, respectively. More mature data, especially from the new FOLFIRI/OSE2101 arm, will provide insight into the incremental benefit of this vaccine-based approach compared with standard continuation maintenance therapy. In an alternative immunomodulatory approach, the toll-like receptor (TLR)-3 agonist rintatolimod [50] is the subject of an early pilot study, whose efficacy outcomes are awaited.

## 4. The Future of PDAC Maintenance Therapy

The concept of maintenance therapy for metastatic PDAC is of considerable and burgeoning interest in the oncologic community but requires additional study, with only a few randomized prospective trials completed to date. As further advances in first-line systemic therapy optimistically lead to deeper and more durable responses with prolonged periods of disease control, we expect a greater number of patients to be candidates for maintenance treatment, highlighting this expanding niche in PDAC treatment as a key opportunity for developing novel therapeutic strategies. 

Table 1 provides a comprehensive list of ongoing trials in the maintenance setting for which results have yet to be reported. As noted in the previous section, the combination of PARP inhibitors with immunotherapy remains an area of particular interest and is the subject of several phase II trials, including a current U.S. cooperative group study from the Southwest Oncology Group (S2001), in which patients with germline *BRCA1/2* mutations who have stable or responding disease after at least 16 weeks of platinum-based chemotherapy are randomized to receive olaparib alone or in combination with the immune checkpoint inhibitor pembrolizumab (NCT04548752). In one case report, a patient demonstrated a radiographic complete response to this olaparib/pembrolizumab combination in the maintenance setting, albeit in the atypical context of harboring both a germline *BRCA1* mutation and an extremely high tumor mutational burden [51]. Other trials are evaluating this same combination among non-biomarker-selected patients (NCT04753879, NCT04666740, and NCT04753879), although the limited efficacy of niraparib/nivolumab reported by Reiss and colleagues in the aforementioned phase Ib/II study raises some doubts as to this particular therapeutic strategy in an unselected cohort. A large effort led by the French PRODIGE-GERCOR cooperative group (the MAZEPPA study) is seeking to personalize maintenance therapy according to tumor-specific molecular features. In this trial (NCT04348045), patients with disease control following 4 months of FOLFIRINOX receive maintenance olaparib if they have a *BRCA1/2* mutation or other evidence of “BRCAness” in their somatic tumor profile; the remaining subjects who do not fit this criteria but whose tumors harbor a *KRAS* mutation are randomized to receive either durvalumab (an anti-PD-L1 monoclonal antibody) plus the MEK inhibitor selumetinib or continued chemotherapy with FOLFIRI.

Other maintenance studies of interest include one evaluating the combination of pembrolizumab with the multikinase inhibitor lenvatinib (NCT04887805); another coupling pembrolizumab with the vitamin D analog paricalcitol (NCT03331562); one employing a combined approach of immune stimulation and CD40 agonism (NCT05484011); one combining the targeting of the T cell immunoreceptor with immunomodulation and the ITIM domain (TIGIT) with immunotherapy and CD40 agonism (NCT05419479); and a dual-immunotherapy strategy combining nivolumab with a novel CXCR1/2 inhibitor (NCT04477343). The authors of this paper are leading a non-immunotherapy-based multicenter phase Ib/randomized II trial in which patients with metastatic PDAC receive capecitabine with or without a histone deacetylase inhibitor in the maintenance setting, following a minimum of 16 weeks of induction FOLFIRINOX (NCT05249101). 

This diverse portfolio of PDAC maintenance trials will hopefully lead to one or multiple new options for a disease currently treated almost exclusively with cytotoxic chemotherapy. However, once mature data from these trials become available, we will need to be careful in how we interpret the results and especially should avoid trying to make inaccurate cross-study comparisons, as differences in a variety of factors will need to be taken into account, including the duration and type of induction chemotherapy allowed for any given trial; the prognostic and predictive implications of biomarkers, if used for study eligibility; and, in randomized trials, the selection of active (e.g., capecitabine) or inactive (e.g., placebo) comparator arms that could influence the perceived effect of the experimental arm(s). 

## 5. Conclusions

The growing interest in maintenance therapy for PDAC is a positive development, as it reflects gradual improvements in first-line systemic therapy over time that now confer durable disease control in increasing numbers of patients with advanced disease. For such individuals, the incorporation of a thoughtful, and possibly tailored, maintenance strategy will ideally optimize progression-free and overall survival while taking into account quality of life considerations that are so important in this non-curative setting. 

Additional prospective randomized clinical trials are clearly needed in this field; ideally, such studies will employ several best practices to facilitate the interpretation of results. First, trials should uniformly report PFS2 and OS data, as these metrics indicate whether any survival benefit was gained by extending first-line therapy with a maintenance strategy versus whether the same effect could be achieved by reserving additional therapies for later lines of treatment at the time of disease progression. Second, patient-reported outcomes will be essential to gauge the impact of maintenance therapy on quality of life and inform patients’ decisions between maintenance and a potential treatment holiday. Third, blood- and tissue-based correlative studies should be embedded within trial design to better discern why some treatments appear effective in the maintenance setting but not for refractory or progressive disease; such understanding may even prompt a revisiting of certain treatments with promising biologic rationale that previously proved ineffective for progressive disease but may find a role in this biologically unique clinical setting of a maximally cytoreduced, responding disease state.

In summary, for patients with metastatic PDAC, treatment decisions often prioritize therapeutic intensity or quality of life to the exclusion of the other. In the future, successful maintenance strategies will ideally make these goals compatible, simultaneously improving patients’ survival and overall well-being.

## Figures and Tables

**Figure 1 cancers-15-03657-f001:**
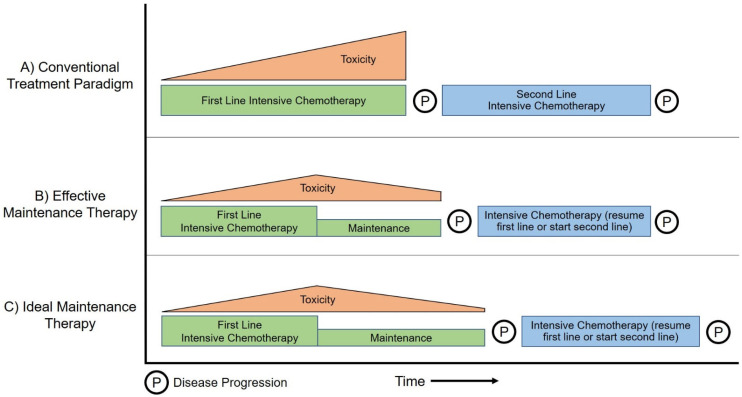
Schematic comparison between the conventional treatment paradigm (**A**) in which first-line chemotherapy is continued until disease progression, effective maintenance therapy (**B**) in which de-escalation of first-line therapy minimizes cumulative toxicity without compromising progression-free survival, and ideal maintenance therapy (**C**) in which de-escalation to maintenance therapy minimizes cumulative toxicity while extending progression-free survival.

**Table 1 cancers-15-03657-t001:** Ongoing trials testing maintenance treatment for pancreatic ductal adenocarcinoma.

National Clinical Trial Identifier	Phase	N	Genetic Subgroup (If Any)	Induction Regimen	Minimum Induction Duration	Maintenance Regimen
NCT02945267	IV	60	-	Gemcitabine + S-1 + Nimotuzumab	Unknown	S-1 + Nimotuzumab vs. S-1 + placebo.
NCT04300114	III	136	Germline *BRCA1/2* or *PALB2* mutation	Platinum-based chemotherapy	16 weeks	Fluzoparib vs. Placebo.
NCT04887805	II	28	-	Chemotherapy	16 weeks	Lenvatinib + Pembrolizumab.
NCT04753879	II	38	-	Gemcitabine + nab-paclitaxel + capecitabine + cisplatin + irinotecan	24 weeks	Olaparib + Pembrolizumab.
NCT03331562	II	24	-	Chemotherapy	Unknown	Pembrolizumab + Paricalcitol vs. Pembrolizumab + Placebo.
NCT04548752	II	88	Germline *BRCA1/2* mutation	Platinum-based chemotherapy	16 weeks	Olaparib + Pembrolizumab vs. Olaparib.
NCT04348045	II	307	-	FOLFIRINOX	16 weeks	Olaparib if *BRCA1/2* mutation. Durvalumab + selumetinib vs. FOLFIRI if somatic *KRAS* mutation.
NCT04666740	II	63	-	Platinum-based chemotherapy	16 weeks	Olaparib + Pembrolizumab.
NCT04390399	II	328	-	Gemcitabine + nab-paclitaxel	Unlimited	SBRT + gemcitabine + nab-paclitaxel vs. SBRT + cyclophosphamide + gemcitabine + nab-paclitaxel + aldoxorubicin + N-803 vs. SBRT + cyclophosphamide + gemcitabine + nab-paclitaxel + aldoxorubicin + N-803 + PD-L1 t-haNK cells.
NCT04753879	II	32	-	Gemcitabine + nab-paclitaxel + Capecitabine + Cisplatin + Irinotecan	Unknown	Olaparib + Pembrolizumab.
NCT05249101	Ib/II	70	-	FOLFIRINOX	16 weeks	Capecitabine + Ivaltinostat vs. Capecitabine.
NCT05419479	Ib/II	46	-	FOLFIRINOX	16 weeks	Domvanalimab + Zimberelimab + APX005M vs. FOLFIRI.
NCT03376659	I/II	8	-	Chemotherapy	8 weeks	MVA-BN-CV301 (prime) + FPV-CV301 (boost) + Durvalumab + Capecitabine.
NCT04627246	Ib	12	-	PEP-DC in combination with either FOLFIRINOX or Gemcitabine + Capecitabine	24 weeks	Nivolumab.
NCT05484011	Ib	30	-	Chemotherapy	16 weeks	Odetiglucan + CDX-1140.
NCT05088889	I	10	-	Chemotherapy	4 cycles	Ipilimumab + Nivolumab + SBRT.
NCT04477343	I	20	-	Chemotherapy	16 weeks	SX-682 + Nivolumab.

SBRT denotes stereotactic body radiation therapy. PD-L1 denotes programmed death ligand 1. t-haNK denotes targeting high-affinity natural killers. PEP-DC denotes personalized peptide dendritic cell vaccine.

## Data Availability

The data can be shared up on request.

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
