# Peer review of "Maintenance Treatment for Metastatic Pancreatic Cancer: Balancing Therapeutic Intensity with Tolerable Toxicity"

_cancers, 2023, doi:10.3390/cancers15143657_

Round 1

Reviewer 1 Report

Authors compiled a manuscript on topic “Maintenance Treatment for Metastatic Pancreatic Cancer: Bal-ancing Therapeutic Intensity with Tolerable Toxicity”. This study focus on the improvement of the maintenance treatment for metastatic pancreatic cancer to improve the quality of life. Authors very well compiled the manuscript and can be published with addressing the following points:

1.     In point 2 i.e., “Continuation Maintenance Therapy”, number of references are very less. Please quote the text with proper reference.

2.     More clinical studies could be added to the manuscript to support the findings.

3.     Authors can draw more figures/flow chart to make the manuscript more presentable.

Author Response

1. In point 2 i.e., “Continuation Maintenance Therapy”, number of references are very less. Please quote the text with proper reference.

We have tried to provide a comprehensive discussion of maintenance therapy in this setting with all published references relevant to this specific topic, including the addition of a couple of additional references in the text as suggested by the second reviewer. It is important to recognize that the overall number of studies published (or presented) in the maintenance setting is relatively modest, as this is a relatively new therapeutic niche that is now being explored. 

2. More clinical studies could be added to the manuscript to support the findings.

In addition to including all relevant studies in the published literature, we have again gone through the clinicaltrials.gov database to try to include as many ongoing clinical studies exploring novel approaches in the maintenance setting in Table 1, and we have added several to this list including:

  • NCT05484011: A Maintenance Therapy Study of Odetiglucan With a CD40 Agonist (CDX-1140) in Patients With Metastatic Pancreatic Ductal Adenocarcinoma
  • NCT04753879: Multi-agent Low Dose Chemotherapy GAX-CI Followed by Olaparib and Pembro in Metastatic Pancreatic Ductal Cancer
  • NCT05419479: Phase Ib/II Open-Label, Multicenter, Randomized Study Evaluating the Safety and Efficacy of 'Switch Maintenance' Combination Immunotherapy Using AB154, AB122, and APX005M in Patients With Metastatic Pancreatic Cancer

3. Authors can draw more figures/flow chart to make the manuscript more presentable.

Thank you for this comment. For space and stylistic reasons, we opted to include one figure designed to help visualize the idea of maintenance therapy for those unfamiliar with the concept; and one table to summarize ongoing studies on this topic. This review article was not designed as a systematic review or meta-analysis, and as such there is no specific flow chart (such as a PRISMA diagram) to include.

Reviewer 2 Report

The article entitled:"Maintenance Treatment for Metastatic Pancreatic Cancer: Balancing Therapeutic Intensity with Tolerable Toxicity" by Evan Walker and Andrew Ko is overall well written, introduce, and conclusions are supported by results. The Overall manuscript is rather interesting; however,  I would suggest some amendments:

1) In point number 2, you should take into account the results of the phase I/II trial (Carratto et al) , pubished in 2022 ASCO annual meeting:

-Phase I/II trial of sequential treatment of nab-paclitaxel in combination with gemcitabine followed by modified FOLFOX chemotherapy in patients with untreated metastatic exocrine pancreatic cancer:

·       DOI: 10.1016/j.ejca.2020.07.035

-In point number 3 “Novel Agents as Maintenance Therapy”. It could be convenient to mention erlotinib, as a failed therapy. The LAP07 clinical trial explored erlotinib with gemcitabine as maintenance in locally advanced PDAC:

DOI: 10.1001/jama.2016.4324

Also,  bevacizumab that has been explored in several tumor types and should be mentioned. Here, I give you some examples of clinical trials that explored it in pancreatic ductal adenocarcinoma:

 DOI: 10.1200/JCO.2010.28.1386

DOI: 10.1200/JCO.2008.20.0238

It is OK

Author Response

  1. In point number 2, you should take into account the results of the phase I/II trial (Carratto et al), published in 2022 ASCO annual meeting :Phase I/II trial of sequential treatment of nab-paclitaxel in combination with gemcitabine followed by modified FOLFOX chemotherapy in patients with untreated metastatic exocrine pancreatic cancer: DOI: 10.1016/j.ejca.2020.07.035”

We greatly appreciate the reviewer’s insight that sequential treatment with gemcitabine-based and fluoropyrimidine-based regimens can be included in this discussion. We have added a new paragraph on Page 4-5 to address this.

  1. In point number 3 “Novel Agents as Maintenance Therapy”. It could be convenient to mention erlotinib, as a failed therapy. The LAP07 clinical trial explored erlotinib with gemcitabine as maintenance in locally advanced PDAC: DOI: 10.1001/jama.2016.4324

We appreciate the reviewer’s insight that the LAP07 trial, while primarily focusing on consolidative radiation for locally advanced disease, did use erlotinib as maintenance therapy. While this manuscript focuses on metastatic disease, we agree that this point should still be included. We have now addressed this in the manuscript with a new paragraph on Page 7.

  1. Also, bevacizumab that has been explored in several tumor types and should be mentioned. Here, I give you some examples of clinical trials that explored it in pancreatic ductal adenocarcinoma: DOI: 10.1200/JCO.2010.28.1386. DOI: 10.1200/JCO.2008.20.0238.

We agree with the reviewer that bevacizumab has been explored in several tumor types as a possible maintenance therapy, a point we acknowledge on Pages 2 and 3. However, this agent has not been evaluated specifically in the maintenance setting for metastatic pancreatic cancer, which is the sole focus of the current manuscript. The studies referenced by the reviewer were studying bevacizumab in combination with gemcitabine or gemcitabine/erlotinib respectively in the front-line setting for advanced pancreatic cancer, not in the maintenance setting. Thus, we do not believe that inclusion of these references would be quite appropriate to include here.